# Assessing population dynamics in the Central Salish Sea, Pacific Northwest Coast of North America

Adam Rorabaugh [1,2]*

1 Washington State Department of Fish and Wildlife, Olympia, WA, United States of America, 2 Department of Archaeology, Simon Fraser University, Burnaby, Canada

* adamnr.2001@gmail.com

## Abstract

Recent developments in radiocarbon dating have enabled archaeologists to re-examine the question of population dynamism in the Salish Sea. This study expands on Taylor and colleagues (2011) using Kernel Density Estimation (KDE) and an expanded data set of 538 radiocarbon dates from academic and cultural resource management literature. The expanded sample suggests a pattern of population growth from 3200–2800 cal BP in coastal Northwestern Washington, with population growth in the San Juan islands during 2600–2200 cal BP. A subsequent decrease in radiocarbon frequencies and large sites suggests shifts in use of the San Juan Islands, followed by peak large-scale occupation from 650–300 cal BP. This pattern is robust whether marine or terrestrial dates are considered. However, marine dates are less sensitive to questions at smaller temporal scales. The broad scale radiocarbon frequency patterns observed are also consistent with those observed in southwest coastal British Columbia (Ritchie et al., 2016; Morin et al., 2018).

**Data Availability Statement:** All relevant data are within the paper and its Supporting Information files.

**Funding:** The author received no specific funding for this work.

## Introduction

There has been renewed interest in examining large radiocarbon datasets to address questions of continuity and settlement dynamics in the Pacific Northwest Coast [1–6]. However, such work south of the Canadian border in the Salish Sea (consisting of the Strait of Georgia, Strait of Juan de Fuca, and Puget Sound) has been limited. The largest published synthetic analysis of coastal dates south of British Columbia has been-the San Juan Archaeological Project (SJAP), which also critically examined potential systematic biases in the use of marine shell dates [7–9]. Expanding on that work, this study assesses radiocarbon ages from archaeological sites from the San Juan Islands, Northwestern Washington state, and the interior of northwestern Washington using updated calibration techniques [10, 11]. To date, no synthetic study of radiocarbon dates to assess population dynamics of the southern Strait of Georgia and the interior of the Cascade mountains has been conducted. This expanded data set from Washington State is compared with broad patterns seen in recent revaluations of population dynamics in British Columbia [2, 3].

**Competing interests:** The author has declared that no competing interests exist.

This study is divided into three regions 1) The San Juan Islands 2) Northwestern Washington 3) Northwestern Washington Interior. These divisions are based on both geography and research history. In terms of research history, the San Juan Islands were subject to early study by Carlson [12, 13] and the subsequent San Juans Archaeological Project (SJAP) conducted under the direction of Dr. Julie K. Stein. Sites from Northwestern Washington include previously recorded sites in Whatcom and Skagit counties with the exception of site 45SK421, which was included in the SJAP. Work at these sites has been a mixture of compliance archaeology from cultural resource management in the region and concerted research by faculty at Western Washington University, including Garland Grabert and Sarah K. Campbell. While data from these sites have contributed to numerous master's theses and studies, this paper is the first concerted effort to synthesize radiocarbon dates from the region. Finally, the divide between interior and coastal sites is 10 km from the shoreline, which separates riverine and montane sites from coastal sites. Sites in the interior were primarily studied by Mierendorf as part of ongoing site inventory and management efforts of the North Cascades National Park.

Coastal sites examined are north of Puget Sound and south of the Canadian Border, an area without a synthesis of the archaeological work of the area outside of the San Juan Islands. The spatial scales used in this analysis are intended to allow for comparison with the British Columbia analyses, and to find a balance between large regional scales that could obscure discontinuities (Fig 1). Similarly, the unit of analysis employed here is larger than specific settlements or households to avoid the detection of localized temporal gaps inconsistent with the larger trends on the landscape [2]. Additionally, this study answers the call for additional comparative analyses by viewing the Salish Sea as a comprehensive cultural area [14].

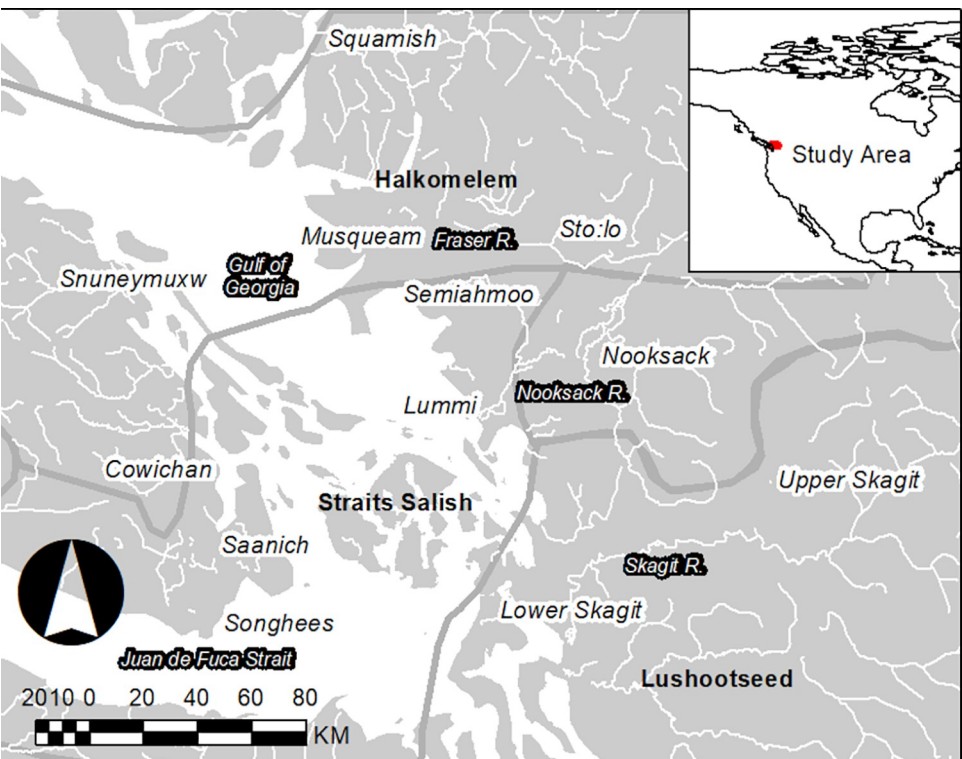

**Fig 1. Study area and ethnohistoric dialect groups (adapted from [26]).**

Another thrust of this paper is to provide a meaningful comparison of marine and terrestrial dates. While some researchers have omitted marine dates and samples with potential marine diet contributions from their analyses due to concerns regarding local marine reservoir effects (Page 142 in [2]) marine shell from shell bearing sites provide direct measures of human activities. This is a result of shell bearing sites formed by the "basketload" (Page 30 in [13]) [15], and shell being utilized for landscaping and monumental construction [16]. As such, despite potential methodological issues with calibrations, shell dates are a crucial line of evidence for examining human occupation and use of coastal areas.

A Kernel Density Estimation (KDE) model is used to account for smaller sample sizes after dividing dates by region, and by terrestrial or marine dates. KDE is a useful approach for datasets where the underlying distribution of events are not well understood (Page 1831 in [17]). KDE has been employed in other regions to determine the underlying structure of radiocarbon datasets [18–25]. Additionally, for clarity of discussion, uncalibrated radiocarbon ages are hereby referred to as "BP" while calibrated radiocarbon ages are "cal BP."

## Regional background- The Salish Sea

The Salish Sea has been a central area for theorizing the emergence of hereditary social inequality in non-state societies, and the management of landscapes and food production by sedentary 'foragers' [27–34]. The regional focus of this study, the San Juan Islands and Northwestern Washington State, is in the traditional areas of the Lummi, Nooksack, Samish, Sauk-Suiattle, Stillaguamish, Swinomish, and Upper Skagit.

Anthropological explanations of hereditary social inequality in the region have traditionally drawn heavily from cultural ecology and have ranged from resource depression to unequal access to resources or charismatic "tectonic" leadership. Demographic models proposed in the region have historically emphasized in situ population growth and population pressure as casual agents for the emergence of hereditary inequality in the region [35]. Regional faunal analyses have increasingly challenged such interpretations by demonstrating continuity in the use of faunal resources [36–39]. The social transformations in the Salish Sea from 3200–1000 cal BP which include a transition from smaller house structures to the emergence of large plank house villages, food storage, resource ownership, and a transition from achieved to hereditary forms of social inequality have been increasingly examined through a lens of historic processes [40–43]. By 2000 cal BP, the archaeological data, specifically the burial record, suggest hereditary forms of social inequality [27, 30, 44–47], resource storage, and large households similar to the ethnographic record [42, 48–52]). Ethnohistoric sources also describe seasonal rounds seen throughout the region [26, 53–59]. Two to four residential moves were typical per year for many Northwest Coast groups (Page 280 in [53]). Kin groups would typically aggregate during the fall and winter months, resulting in villages consisting of multiple households [55, 56, 59]. These winter village groups would mass harvest seasonally available resources including geophytes, terrestrial and marine mammals, shellfish, and anadromous fish for winter storage. During spring and summer months, winter villages dispersed into smaller household groups for harvesting seasonal resources that were owned by extended kin groups [60, 61]. The recruitment and maintenance of household labor in winter village groups was also key, as individuals could express their autonomy by "voting with their feet" and changing their residence [48]. This resulted in emergent fission and fusion dynamics within winter village populations.

While past population pressure models are not supported by the faunal record [35] questions remain about overall population growth patterns and variation throughout the Salish Sea through the Holocene. While pre-contact demographic data are sparse, there have been

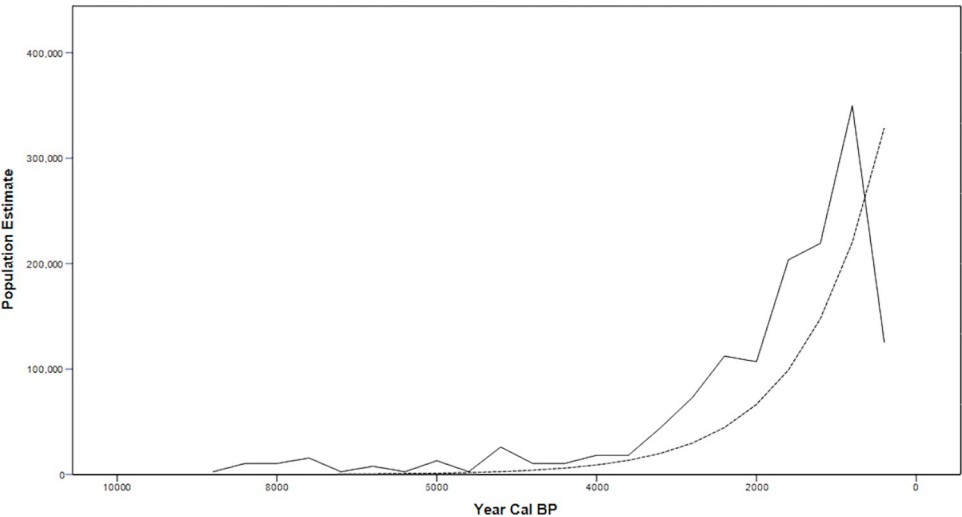

**Fig 2. Salish Sea Holocene population estimates.** Solid Line- Ames and Maschner [54] curve with Boyd [63] Projected Population Count Dotted Line- Croes and Hackenberger [35] model (Adapted from Rorabaugh [64]).

seminal efforts examining in situ population growth through the Holocene are of note. First, Croes and Hackenberger [35] modeled population growth through the Holocene with an annual increase of 0.1% (Fig 2). Second, Boyd [62, 63] constructed 18th and 19th century population histories, albeit highly conservative population estimates, and a definitive epidemic history of the region. Finally, Ames and Maschner [54] constructed a population curve based on summed radiocarbon frequencies. a research program followed in the past twenty years with higher regional detail in British Columbia.

These recent efforts in coastal British Columbia have provided a wealth of data for sites within the northern Salish Sea [1, 2]. However, south of the Canadian border, recent synthetic studies have been more limited in scope and address specific questions such as the efficacy of specific dating methods [5]. This study aims to contribute to this known research gap by providing an updated synthesis of dates south of the Canadian border, specifically Northwestern Washington and its interior which have not yet been subject to a synthetic analysis of radiocarbon dates. This analysis is a crucial step for basic research that is required to further address these aforementioned key anthropological questions in the future by using other lines of evidence than the faunal record.

## Research questions

As an expansion of the work done for the SJAP, this study focuses on whether the patterns in radiocarbon date frequencies in coastal northwest Washington and the interior of northwest Washington better reflect trends in the San Juan Islands, Fraser Delta, or Fraser Valley. Some researchers [65], based on culture-historic sequence building, have argued that the Fraser Delta served as a regional core, with the emergence of large plank house villages, hereditary social inequality, observed in the ethnohistoric period "the Developed Northwest Coast Pattern" emerging first in the Fraser Delta and subsequently spreading to the rest of the Salish Sea. Setting aside other critiques of [14, 66] that study if this is the case, it would be expected that overall population trends in this study area would differ from those in British Columbia in either timing or intensity of occupation. As marine dates have been described as unreliable by

some researchers [1], whether marine dates present systematic biases unaccounted for by local reservoir curves is also assessed by comparing terrestrial and marine date datasets.

Another concern raised by researchers regarding the accuracy of both marine shell dates [67] and assessing population dynamics of coastal sites has been the potential impact of coastal erosion. While coastal erosion has impacted shell bearing sites, this tends to be the result of periods of stability punctuated by decadal mass wasting events [68]. Long fetch shorelines with steep bathymetry also tend to be more impacted by these mass wasting events [69]. While more recently deposited sites are more abundant in these high erosional risk areas of the San Juan Islands, post site formation processes do not fully explain the patterns in site sizes and frequencies (Page 301 in [7]). As such, despite the potential for differential erosion processes, they are not expected to cause significant impacts on the radiocarbon frequency trends between regions.

## Methodology

### Database

Radiocarbon dates were compiled from the Washington Information System for Architectural and Archaeological Records Data (WISAARD) and on file with the department of anthropology at Western Washington University. These include radiocarbon dates from published literature, archaeological survey reports, unpublished master's theses, doctoral dissertations, and dates submitted to the Department of Archaeology and Historic Preservation (DAHP). The compilation of dates spans research from 1969 to 2022 and includes all submitted gray literature to the Washington State Historic Preservation Office. While the geographic focus of the study is Washington State, for some examined sites, key associated records were on file with the British Columbia Heritage Branch and had dates in the Canadian Archaeological Radiocarbon Database (CARD). The gray literature review of this study includes dates not present in the Kelly and colleagues [6] and Hutchinson [70] databases. Only radiocarbon ages that could be tied to cultural deposits were included, and dates stated by report authors as erroneous were excluded. As all cultural deposits with dates were included, the types of sites in this study range from temporary camps to large plank house villages. The emphasis of this study is regional demographic trends, not specific settlement or site type analyses. An aim of this study was also to critically assess uncertainties in marine dates. Consequently, shell and bone dates with marine diet contributions were also included. The sample consists of 538 radiocarbon dates (N = 317 terrestrial dates, N = 221 marine dates) from 142 sites. Sites were classified into three spatial regions San Juan sites (including 45SK421 for consistency with previous SJAP research), NW Washington, and NW Washington Interior (Fig 3). While as Prentiss and colleagues [71] have noted that legacy radiocarbon data are not always ideal, this study is intended to compile and provide access to "gray literature" dates from a portion of the Salish Sea with considerable archaeological research from the cultural resource management sector which has been underrepresented in discussions of the Northwest Coast.

### Calibration procedure

Radiocarbon ages were calibrated with OxCal 4.3.2 [72] using the IntCal20 atmospheric curve [11] and the Marine20 global marine reservoir curve [10]. Original lab reporting of dates were re-examined and δ13/12 fractionation corrections [73, 74] were applied when such data were available. When d13C data were not available, values of -25 for charcoal and -1.53 for shell, based on mean marine shell sample d13C data, were assigned, adapting Osterkamp and colleagues' procedures to marine dates (Page 550 in [75]). Deo and colleagues' [9] and Daniels' [8] local marine reservoir curves were applied to marine dates from the San Juans and NW

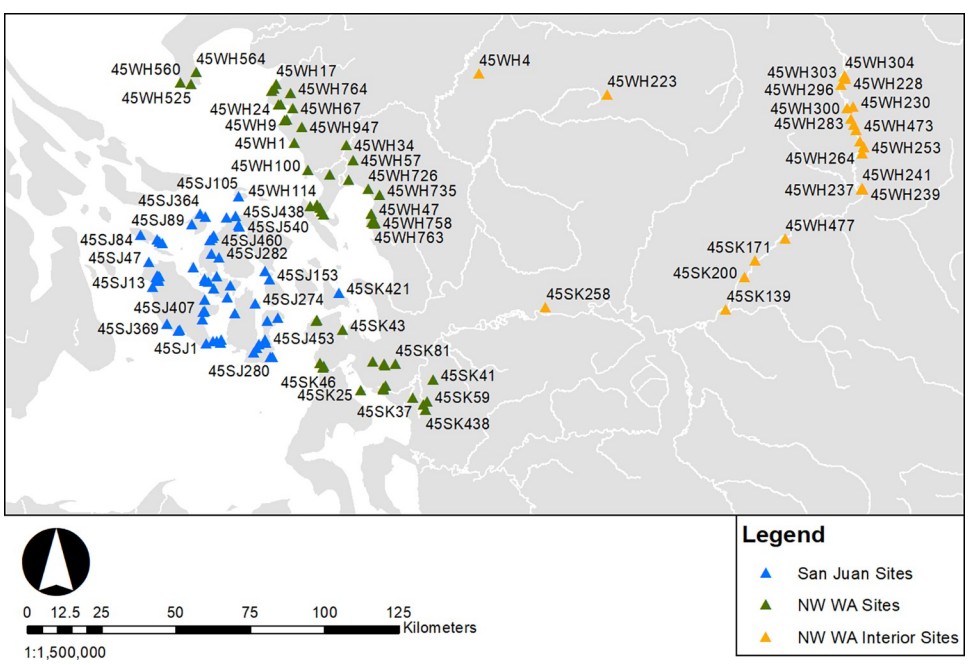

**Fig 3. Examined sites and regional classification.**

Washington sites. The protocols used are in line with the recommendations of Hadden and colleagues [67].

## Kernel Density Estimation

This study primarily employed Kernel Density Estimation (KDE) instead of a Summed Probability Distribution (SPD) function approach. While SPDs have been commonly employed for regional [1, 2, 76–80] and village or household scale analyses in the Pacific Northwest [81–85], KDE is a useful method for datasets where the underlying distribution of events is poorly understood (Page 1831 in [17]) as may be the case with marine dates. While not used extensively in the Pacific Northwest (c.f. [23]), KDE and composite KDE have been successfully used to determine the underlying structure of radiocarbon datasets [18–22, 24, 25, 71].

KDE is a non-parametric method and does not require assumptions of a particular underlying distribution, unlike parametric Bayesian approaches (Page 1831 in [17]), and is intended to mitigate uncertainty from sampling error. Additionally, the ability to examine variability in the distributions during Monte Carlo Markov Chain (MCMC) analysis also is an advantage over alternative approaches that present a single solution for a dataset. 500,000 MCMC replicates were used to generate the KDE distribution.

## Results

### San Juan Islands

The marine dates collected from the San Juan Islands (Fig 4) exhibit an early peak at 2600–2200 cal BP (Fig 5), not apparent in the general regional demographic model (Fig 2). While the bandwidth upper limit is wider with marine dates as a result of higher error terms, this does not appear to obscure structure. However, the terrestrial, and combined terrestrial- marine dates show consistent patterns with more pronounced peaks in the combined analysis due to an increased sample size. In the combined sample two peaks are evident; an earlier one at

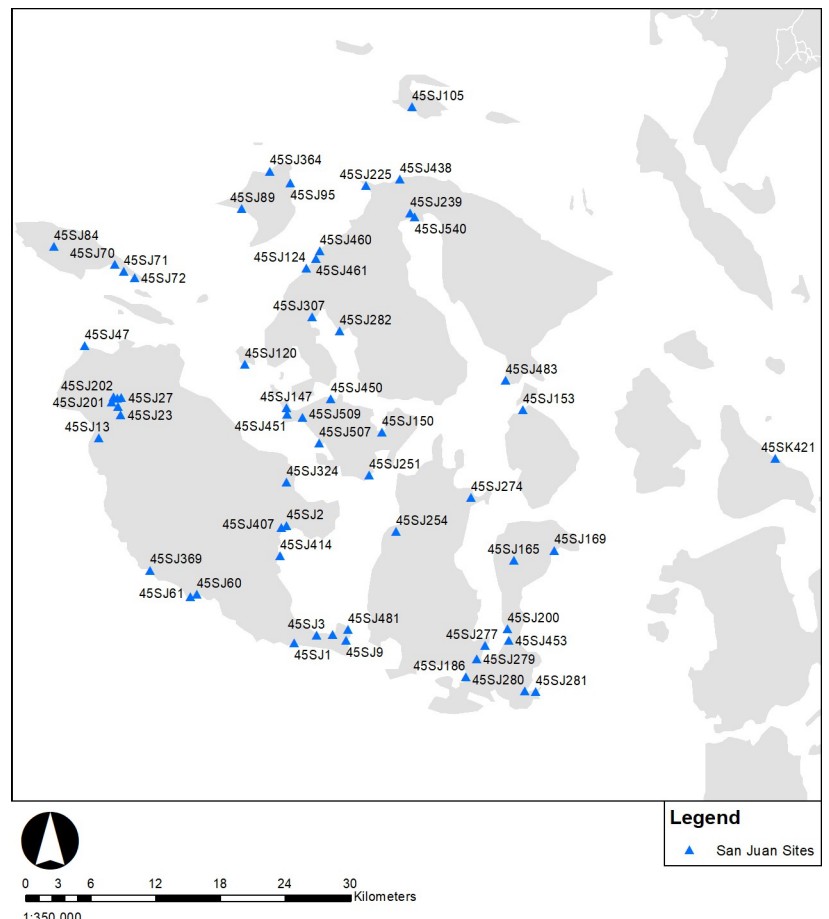

**Fig 4. San Juan Island sites (N = 65).**

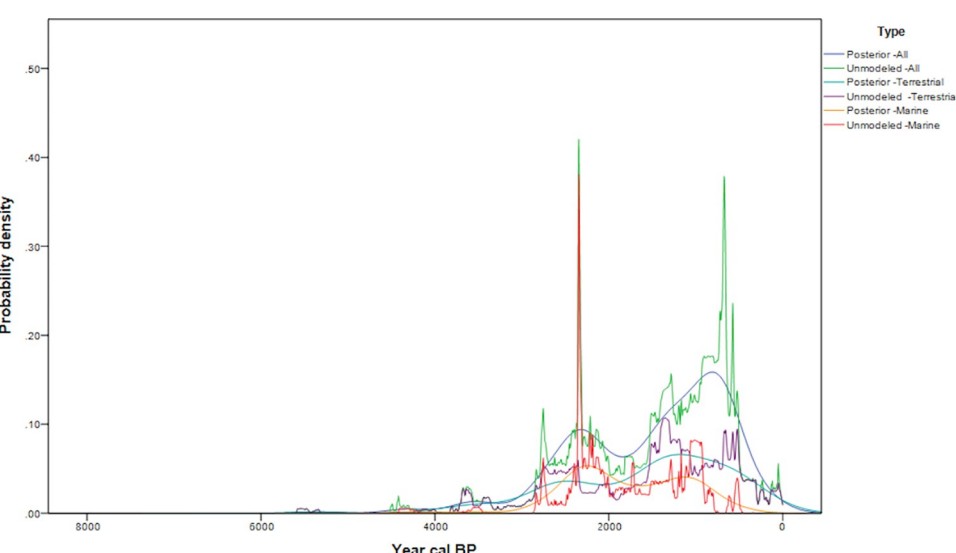

**Fig 5. San Juan Islands date probability densities and model (marine dates N = 134, terrestrial dates N = 144, marine and terrestrial dates N = 278).**

2600–2200 cal BP, and a second at 650–300 cal BP. The differences between the terrestrial, combined, and marine date samples may be due to biases from reservoir effects resulting in older estimates, or due to research history from large sites with large shell date samples and few charcoal or bone dates. Based on the use of local reservoir corrections for the region for shell dates, it is more likely to be a systematic bias from site research history.

## Northwest Washington

With the Northwest Washington sample (Fig 6), marine dates exhibit an earlier peak at 3200–2800 cal BP (Fig 7), also not apparent in the general regional model (Fig 2) and appear to have wider bandwidth due to higher error terms than the terrestrial dates, but its structure is not

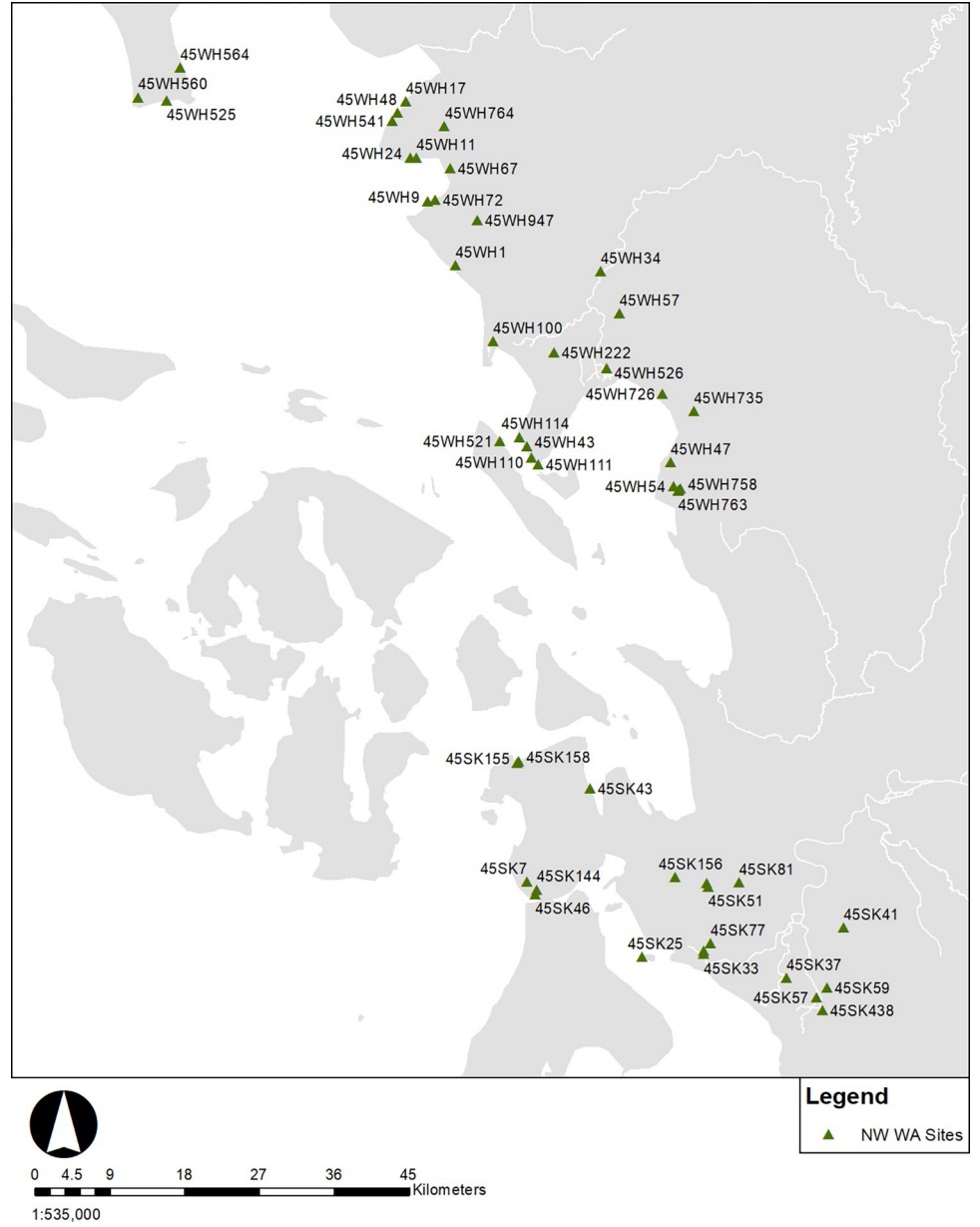

**Fig 6. Northwest Washington sites (N = 55).**

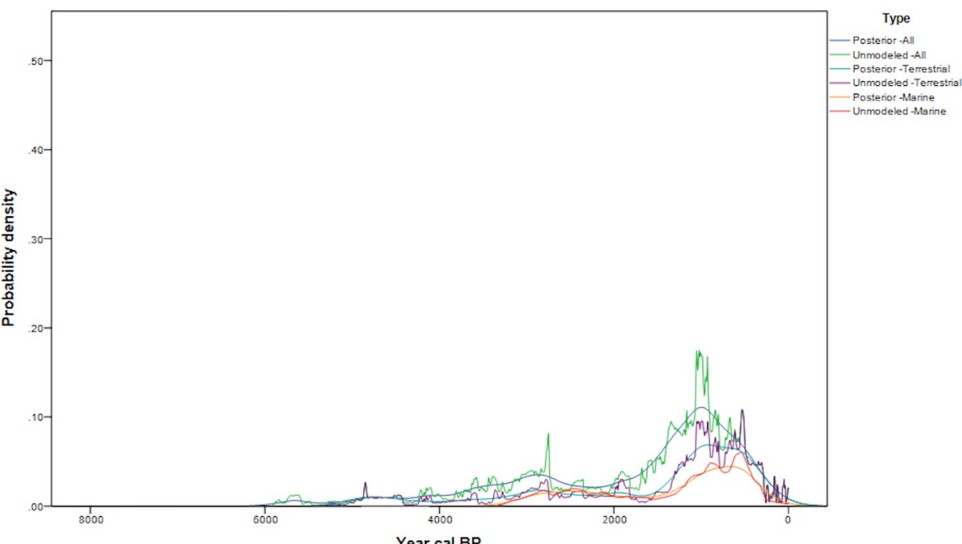

**Fig 7. Northwest Washington date probability densities and model (marine dates N = 77, terrestrial dates N = 109, marine and terrestrial dates N = 186).**

obscured. As with the San Juan sample, the combined terrestrial and marine data set has more pronounced peaks due to an increased sample size. In contrast to the San Juan Islands sample, the terrestrial and marine dates show more consistent broad patterns. These marine dates also use local marine reservoir corrections, and the research histories of several sites specifically focused on providing balanced marine and terrestrial dates in their sampling strategies (ex. 45WH1, 45WH17). This again strongly suggests that site research history more than systematic bias is affecting the reliability of marine shell dates beyond their having wider error terms. The terrestrial sample exhibits several periods with probability peaks at 5100–4900 cal BP, 3200–2800 cal BP, 2200–2400 cal BP, and 1100–800 cal BP. The combined sample exhibits two peaks, the earliest at 3200–2800 cal BP and later at 1100–900 cal BP, which may suggest finer structure being obscured by marine dates.

## Northwest Washington interior

The northwest Washington interior sample (Fig 8) consists of terrestrial dates. Four pronounced shifts are apparent in the data (Fig 9); the first is a peak at 6000–5500 cal BP, followed by monotonic increase after 4500 cal BP with plateaus at 4500–4000 cal BP 1250–750 cal BP and 750–300 cal BP. The absence of periods with decreased 14C age probabilities after 4500 cal BP is a contrast to terrestrial date samples of the other regions.

## Comparing posterior probabilities between regions

Both the marine and terrestrial datasets suggest that patterns of increased frequencies of radiocarbon samples from 3200–2800 cal BP in coastal Northwestern Washington, with increased frequencies in the San Juan islands during 2600–2200 cal BP (Fig 10). A subsequent decrease in radiocarbon frequencies and large sites suggests shifts in use of the Islands [7], followed by a peak in large-scale occupation from 650–300 cal BP. This dynamism is in contrast to past regional demographic models (Fig 2). The curves in Northwest Washington and the Northwest Washington Interior follow the pattern seen in the San Juans with a high frequency of dates from 650–300 cal BP. The lower posterior probabilities observed overall in the Northwest

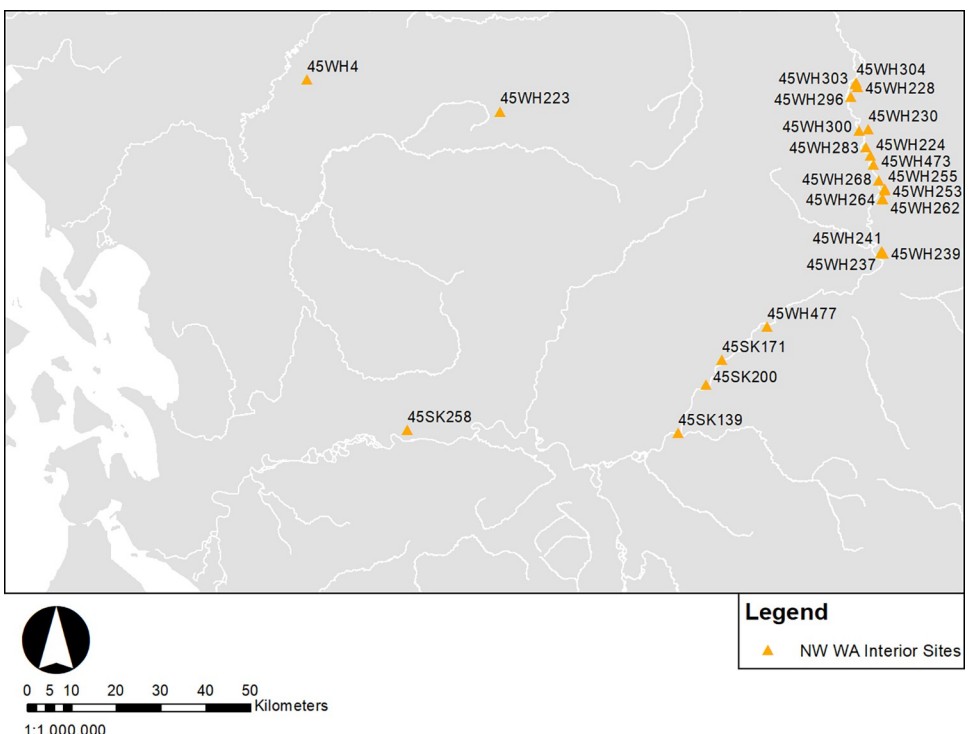

**Fig 8. Northwest Washington interior sites (N = 22).**

Washington Interior sites likely reflect less intensive land use, in part due to seasonal accessibility restrictions in the mountains and foothills. The absence of a decrease in 14C age probabilities after 4500 BP may be a result of these sites being tied to seasonal rounds and as such reflect aggregated use by the kin networks of the islands and mainland coast. Additionally, the monotonic increase of interior date frequencies after 2000 cal BP suggests increased settlement and utilization of the interior shortly proceeding the shift in island use.

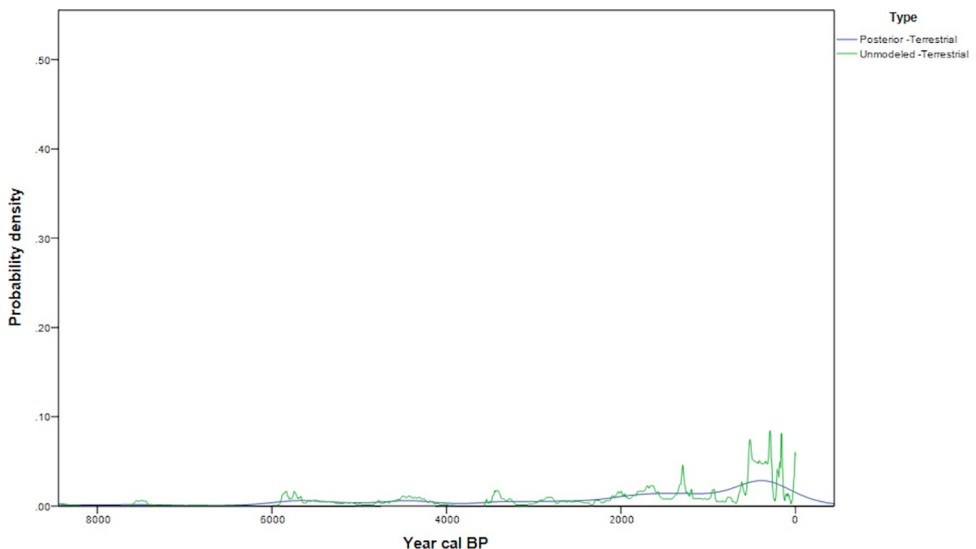

**Fig 9. Northwest Washington interior date probability densities and model (N = 64).**

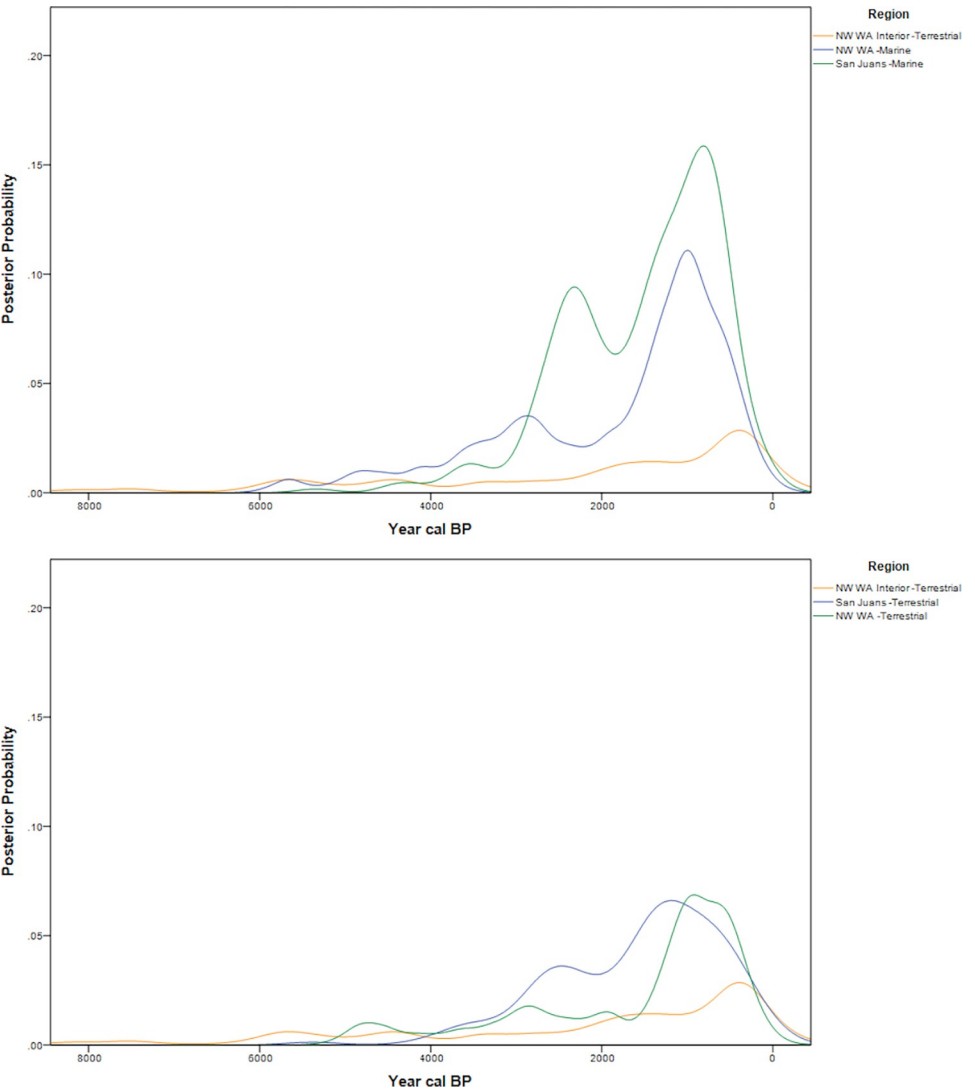

**Fig 10.** Posterior probabilities (Top: Marine+Terrestrial, Bottom: Terrestrial Only).

## Discussion

The findings of this study are best viewed as working hypotheses. While the SJAP provided a wealth of data for the San Juan Islands, this study is the first synthesis of radiocarbon dates from northwestern Washington state to provide comparison with the SJAP, and the research conducted in the Gulf Islands, Fraser Delta, and Fraser River. Overall, these patterns suggest the permeable nature of Coast Salish territorial boundaries through time [86, 87], with territoriality based on practices of kinship, travel, and inheritance of title and corporeal and non-corporeal resources. As researchers increasingly move from unicasual models of population pressure resulting in resource storage and the management of resources as a cause for the emergence of hereditary forms of social inequality, the results of studies such as this provide the basic groundwork for more refined faunal analyses and higher resolution investigations of material culture that can more directly address such key anthropological questions.

Both the combined and terrestrial datasets suggest a pattern of continuing in situ population growth from 3200–2800 cal BP in coastal Northwestern Washington, and population

growth in the islands during 2600–2200 cal BP. Based on faunal data [37, 88] it is too simplistic to state that these regional differences correspond with periods of resource intensification either driving or caused by population pressure but shifts in affinal kin networks and the fission-fusion dynamics of villages may all play roles. However, the population shifts in seen in the data may reflect Lummi oral traditions recounted to Suttles [89]. In these traditions, the first man *sweta'n* came down from heaven to a place on the north end of San Juan Island, which Suttles viewed as most likely to be modern Garrison Bay. An ancestral village off of the bay *Pe'pi'ow'elh* [90] is archaeologically represented by site 45SJ24 was occupied until 1859 when British troops established a military camp and dismantled the plank house during the Pig War [91]. *Sweta'n* became the ancestor of the *tale'quamis*, which Suttles considered to have contributed to the ancestry of the Lummi and likely also contributed to the ancestry of the Saanich and Songhees of southeastern Vancouver Island. Similarly, the placename for San Juan Island, *lháqemesh*, means "to appear or come down" [90]. The teachings of these oral traditions are consistent with the initial trends of in situ population growth on the islands. Brown [3] has also argued that large shell bearing sites in the islands, some of which are still undated, reflect cultural keystone places with deep histories.

A subsequent decrease in radiocarbon frequencies and large sites suggest shifts in the use of the San Juan Islands, including the aforementioned keystone places, followed by peak large-scale occupation in the San Juan Islands from 650–300 cal BP. The Northwestern Washington and Northwestern Washington Interior curves follow patterns seen in San Juan Islands with a high frequency of 650–300 cal BP dates.

The marine dates for the San Juan Islands show a higher increase in 2600–2000 cal BP, followed by a smaller peak at 650–300 cal BP. The disparity between terrestrial and marine dates observed in the San Juan Islands sample is not present for the Northwest Washington marine and terrestrial dates which use the same local marine reservoir corrections. The differences between the terrestrial and marine dates seen in the San Juan Island sample may be attributable to specific research histories of sites and sampling in San Juan Islands.

As noted earlier, these data may, in part, reflect oral traditions. Oral traditions [89, 92] document the Lummi moving from villages on the north end of San Juan Islands, the east and west sounds of Orcas Island, to the mouth of the Nooksack River. Suttles [89] believed that the story was a recent (post 1850) event, but Curtis [93] noted that based on genealogies, this population movement may have not occurred until circa 1725.

The overall pattern of increased date frequencies from 800–600 cal BP for the Fraser Valley noted by Ritchie and colleagues (Page 149 in [1]) is apparent in this dataset. The same general population trends present in this sample suggest that interpretations emphasizing the Fraser Delta as a core and other regions of the Salish Sea as a periphery are too simplistic. Instead, southern Salish Sea demographic trends can be best described as a combination of in situ population growth with historical continuity. Additionally, these trends also likely reflect village fission-fusion dynamics in line with the permeability and autonomy observed in the ethnohistoric period. Averaging all regions with current data also yields monotonically increasing curves consistent with past regional demographic models (Fig 2). This highlights the need for multiscalar analyses to examine broad scale trends in the demographics of the Salish Sea and Fraser Valley, but also that data can capture demographic patterns at regional scales. Comparing datasets both in the United States and Canada is crucial for having a complete picture of precontact Coast Salish population dynamics and the impact of the border on patterning specific research histories in Washington and British Columbia- which have obscured larger regional patterns in the past that cannot be understated. The wider error terms with marine dates, and the need for additional local reservoir corrections should also not dismiss their utility for addressing specific questions. Continuing to develop local marine reservoir curves with

paired marine and terrestrial dates should be a priority, even if these dates are best suited to be used in larger temporal analysis scales. This also highlights the utility of marine shell dates for regional or mesoscale analyses provided that sufficient information on radiocarbon reservoir effects are available.

The decrease in frequency in radiocarbon ages from 500 BP on is also observed in other studies throughout the Pacific Northwest [1, 2, 23, 76, 79]. A constellation of factors contributes to this pattern, including sampling biases in not dating late period archaeological sites and historic sites, impact of modern development on later sites, and the demographic collapse following the indirect and direct contact with Euroamericans [62, 63, 94, 95]. Although there are fewer dates, the historic period has a well-documented dynamic social landscape [96–98].

Areas for future work expanding from this dataset include settlement pattern analyses of sites in Northwestern Washington to compare with the demographic trends presented here and the results of the SJAP. Additionally, a Bayesian approach using finite mixture models (FMM) of sites from the Northwestern Washington and San Juan Islands sample with well-defined priors for site components is another potential direction [99]. While an FMM study of individual sites is more in line with previous site-level radiocarbon analyses previously done in the region, it can provide new insights to specific site histories. FMM is also suitable for modeling regional demographic and settlement pattern dynamics.

Based on the overall results of this study the following key points may be drawn:

1. The overall population growth trends in the San Juan Islands and Northwest Washington are consistent with those in the Fraser Delta from 800–600 cal BP, suggesting in situ population growth and long term participation in large, permeable, kin networks throughout the Salish Sea.

2. Population trends in the San Juan Islands likely reflect settlement patterns discussed in Lummi Oral traditions including initial settlement of and movement from the islands.

3. The combined marine and terrestrial sample lack the fine distributional structures seen in the terrestrial-only sample, but broad patterns are consistent between the two. This suggests that marine dates are useful for mesoscale and larger scale synthetic studies of demographic trends.

4. While shell date error terms may obscure fine structure, reservoir corrections themselves may not be problematic.

This study also highlights the need to synthesize data from the "gray" literature, as a substantial amount of data collected in the cultural resource management literature has been underutilized to examine broad scale patterns in the southern Salish Sea. By synthesizing the data from these reports, this study provides a steppingstone to larger synthetic databases [6, 70]. In addition, his study also highlighted the regional variability in population dynamics, and the need to account for shifts in population structure through time. The shifts observed in radiocarbon frequency data appear to be best interpreted through the lens of the permeability of traditional Coast Salish social boundaries through time, which has broader implications for interpreting the population dynamics of sedentary and semi-sedentary hunter-gatherer-fishers.

## Supporting information

**S1 Appendix. Radiocarbon data.**
(DOCX)

## Acknowledgments

I thank Amanda Taylor, James Brown, and Anna Coon for their comments and feedback on different versions of this manuscript.

No permits were required for the described study, which complied with all relevant regulations.

## Author Contributions

**Conceptualization:** Adam Rorabaugh.

**Data curation:** Adam Rorabaugh.

**Formal analysis:** Adam Rorabaugh.

**Investigation:** Adam Rorabaugh.

**Methodology:** Adam Rorabaugh.

**Writing – original draft:** Adam Rorabaugh.

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
