## [Decision Letter · Decision Letter 0]

19 Oct 2022

PONE-D-22-25181Assessing Settlement Dynamics in the Central Salish Sea, a Bayesian ApproachPLOS ONE

Dear Dr. Rorabaugh,

Thank you for submitting your manuscript to PLOS ONE. After careful consideration, we feel that it has merit but does not fully meet PLOS ONE’s publication criteria as it currently stands. Therefore, we invite you to submit a revised version of the manuscript that addresses the points raised during the review process. Both reviewers indicate that your work has promise to be an important contribution to understanding demographic trends in your study area. However, they also indicate the need for fairly substantial revisions and additions. Reviewer 1 indicates the need for much more contextual information. I agree. When I first read your manuscript, I thought it read like a section/chapter in a much larger report rather than a stand alone journal article. Reviewer 2 indicates the need for corrections in your discussion of kernel density estimates. As implemented in OxCal KDEs are not strictly Bayesian: "Because the KDE method is in itself frequentist, this is not a purely Bayesian approach" (Bronk Ramsey 2017:1819). To better place KDEs in context, I suggest that you read: Crema, Enrico R. "Statistical inference of prehistoric demography from frequency distributions of radiocarbon dates: a review and a guide for the perplexed." *Journal of Archaeological Method and Theory* (2022): 1-32. As Reviewer 2 indicates there are Bayesian approaches to analyzing compiled radiocarbon dates for demographic modeling. Baydem is one possibility, but you do not need to do so for this paper. You do need to correct the sentence on assumptions about underlying distributions. Finally the paper needs a careful proofread. As both reviewers indicate there are partial sentences throughout.

We look forward to receiving your revised manuscript.

Kind regards,

John P. Hart, Ph.D.

Academic Editor

PLOS ONE

Journal Requirements:

2. In your manuscript, please provide additional information regarding the specimens used in your study. Ensure that you have reported specimen numbers and complete repository information, including museum name and geographic location.

"All necessary permits were obtained for the described study, which complied with all relevant regulations."

"No permits were required for the described study, which complied with all relevant regulations."

For more information on PLOS ONE's requirements for paleontology and archaeology research, see https://journals.plos.org/plosone/s/submission-guidelines#loc-paleontology-and-archaeology-research.

3. We note that you have referenced Daniels (2009), Graesch (2006), Palmer (2015), Taylor (2012), Tierney (2012) which has currently not yet been accepted for publication. Please remove this from your References and amend this to state in the body of your manuscript: (ie “Daniels et al. [Unpublished]”) as detailed online in our guide for authors

http://journals.plos.org/plosone/s/submission-guidelines#loc-reference-style "

4. Please ensure that you refer to Figure 10 in your text as, if accepted, production will need this reference to link the reader to the figure.

5. We note that Figures 1,2,3,7, and 11 in your submission contain map images which may be copyrighted. All PLOS content is published under the Creative Commons Attribution License (CC BY 4.0), which means that the manuscript, images, and Supporting Information files will be freely available online, and any third party is permitted to access, download, copy, distribute, and use these materials in any way, even commercially, with proper attribution. For these reasons, we cannot publish previously copyrighted maps or satellite images created using proprietary data, such as Google software (Google Maps, Street View, and Earth). For more information, see our copyright guidelines: http://journals.plos.org/plosone/s/licenses-and-copyright.

a. You may seek permission from the original copyright holder of Figures 1,2,3,7, and 11 to publish the content specifically under the CC BY 4.0 license.   

Reviewers' comments:

Reviewer's Responses to Questions

**Comments to the Author**

1. Is the manuscript technically sound, and do the data support the conclusions?

Reviewer #1: Partly

Reviewer #2: Yes

2. Has the statistical analysis been performed appropriately and rigorously? 

Reviewer #1: I Don't Know

Reviewer #2: Yes

3. Have the authors made all data underlying the findings in their manuscript fully available?

Reviewer #1: Yes

Reviewer #2: Yes

4. Is the manuscript presented in an intelligible fashion and written in standard English?

Reviewer #1: Yes

Reviewer #2: Yes

5. Review Comments to the Author

Reviewer #1: The author has compiled 538 radiocarbon dates from 142 archaeological sites across northwestern Washington State. The dates are grouped into three regions in order to compare demographic trends between them and also with other demographic models existing to the north, in British Columbia, Canada. Utilizing and comparing marine shell dates was also an objective of this study.

The compilation and analysis of these radiocarbon dates has the potential to be a significant contribution, but this study, as is, provides only a cursory and uncritical summary of the data, description of the results, and discussion of demographic trends. More contextual information is necessary for anything beyond the most basic modelling of demography. As it is, a reader would have a challenging time trying to evaluate the appropriateness of the research questions or the validity of the interpretations because no background information in offered on the nature of the c14 dated sites, or the known settlement history of the region. Readers unfamiliar with the region would reasonably wonder how the analysis of the compiled c14 dates relates to earlier research in the region, and advances understanding of key anthropological or historical questions.

Another concern with this study is that there is no effort to look at the settlement patterning over time, to detect if the types of sites or the distribution of sites changed in relation to one another, the environment, new technologies and settlement forms/size, or demographic trends. There is no discussion of factors that might help to explain some of the modelled ups and down in the KDE.

I am not qualified to determine if the marine reservoir corrections have been applied appropriately, but more details would be helpful, and critical analysis of the methods should also be provided considering the ongoing uncertainty and regional variability with using shell dates.

The KDE seem to indicate some interesting trends that are generally consistent with those of other studies, but again, there is not sufficiently detailed results to know what the significance of this might be. Where the models indicate differing trends, there is also not discussion of why this might be.

This manuscript contains many sentence fragments (including the first line of the introduction) and some repetition. Other sections are too brief to be clear to the reader. A background section that might provide useful context for interpretations and discussion is missing.

Altogether, I’d like to see a research paper along these lines published, but would expect it to be thoroughly re-worked, from the intro to the conclusion.

Reviewer #2: This paper updates Taylor et al.'s 2011 paper, exploring population growth dynamics of the Pacific Northwest using the temporal distributions of radiocarbon datasets as proxies for the increasing and decreasing abundance of the archaeological record over time, i.e. the so-called "dates as data" approach. The present manuscript expands on Taylor et al.'s previous work in two noteworthy ways: first, it expands the geographic scope and consequently the sample size of the radiocarbon datasets used by Taylor et al., including adding inland sites from Washington State. Second, it replaces the summed probability distribution (SPD) approach used by Taylor et al. for describing temporal distributions of data points with a kernel density estimation (KDE) approach. Specifically, the author applies the KDE functions introduced to OxCal by Bronk Ramsey (2017), which have the appeal of being (relatively) easily applied by OxCal users without requiring any extensive programming on the user's part.

In general, I think this manuscript makes a novel and nontrivial contribution to our demographic-archaeological knowledge of the Pacific Northwest, and I do hope that it will be published for this reason. I also think that OxCal's KDE functionality supersedes both the SPD approach and other "composite kernel density estimation" alternatives suggested by others reviewed in the submission (e.g. Brown 2017). There is one alternative approach on offer---finite mixture modeling---that is founded on even strong theoretical and methodological principles than OxCal's KDE approach, which I discuss further below, but I will leave it to the author to decide whether the added effort to pursue finite mixture models over a KDE models is worth it; OxCal's KDE functions are still great tools.

One thing I would strongly recommend is that the author give the manuscript a new round of revisions with fresh eyes especially for broken grammar. Most of the paper is clearly written, but I did encounter a handful of sentence fragments, primarily sentence subjects lacking their predicates. Make sure every sentence ends.

Grammar aside, I do have one important critique of the manuscript that I think will need to be addressed before it is published: the author's characterization of kernel density estimation (KDE) as a Bayesian method is incorrect. In fact, KDE is not readily available to standard likelihood based approaches to model-fitting---whether maximum likelihood estimation (MLE; Silverman 1986: 25-26, 110-119; Simonoff 1996: 64-70) or Bayesian inference (Bouveyron et al. 2019: 92-93)---because tuning the KDE model's single free parameter ("bandwidth") in a likelihood-based way leads to the statistical problem of degeneracy: the likelihood function approaches infinity as the strictly positive bandwidth parameter approaches 0. In the case of KDE and other finite mixture models (FMMs), this degeneracy problem sometimes goes under the label "the collapsing variance problem" (Murphy 2012: 357-359), and it is unavoidable in the case of KDE. While it is true in general that OxCal comprises a diverse set of tools for radiocarbon data analysis that are Bayesian in a strict sense, this is not the case for OxCal's recently added KDE operations, a fact about which Bronk Ramsey (2017) is quite explicit. Instead, in order to operationalize KDE, OxCal's almost-but-not-quite-Bayesian approach replaces the proper likelihood function required by Bayes' Rule with a leave-one-out predictive probability function (Bronk Ramsey 2017: Eqs. 5 and 9). (A similar approach has been applied to MLE approaches to KDE; Silverman 1986: 52-55.) On one hand, this substitution "works" because the out-of-sample predictive probability function behaves in a manner similar to likelihood functions, increasing and decreasing over the parameter space depending on how suitable different parameter values are for the data under analysis. On the other hand, these out-of-sample predictive probabilities still should not be mistaken for likelihoods in the strict sense, so their substitution is of questionable theoretical merit. In my opinion, this choice is especially problematic for Bayesian inference because this substitution sacrifices the logical axioms that give this inferential framework, especially Bayes' Rule, its principled foundations. Nevertheless, a certain amount of pragmatic justification does exist for their use as a hack when proper likelihoods just won't work (Silverman 1986: 52-55), so I wouldn't insist that we abandon OxCal's KDE functions altogether. Even so, they are not Bayesian.

In any case, the author will need to make one of two changes to address and resolve this issue:

(1) The simpler solution would be editorial: remove or qualify any characterization of KDE as Bayesian, including in the paper title, key words, abstract, and main body of the text. This would require no new analysis on the author's part but merely a more accurate characterization of what the author has already done.

(2) Alternatively, if the author really wishes to do something that is truly Bayesian, he should consider a new round of analysis, replacing his KDE models with FMMs that can be approached in a Bayesian way. As it turns out, KDE models are an extreme case of FMMs, in which there are as many mixture components as there are data points and in which all kernels or mixture components share the same bandwidth. However, unlike KDE models specifically, FMMs more generally *are* available to Bayesian inference in the strict sense, particularly if there are fewer kernels/mixture components than there are data points. The collapsing variance problem mentioned earlier is still a possibility particularly for FMMs with large numbers of mixture components, but this problem can be mitigated in a Bayesian framework using regularizing priors (Fraley and Raftery 2007; Bouveyron 2019: 92-95) and/or posterior maximization rather than likelihood maximization (Murphy 2012: 357-359). Additionally, using FMMs with relatively few mixture components usually out-performs KDEs when applied as density estimates to the same data, particularly if different kernels/mixture components are allowed to have different bandwidths (Fraley and Raftery 2002; Gelman et al. 2013: 535). Unfortunately, OxCal has not yet implemented functions for finite mixture modeling. To my knowledge, the only prepackaged program capable of fitting such models is the 'baydem' package for R described by Price et al. (2021). If the author wishes to apply a truly Bayesian approach to the task at hand and is knowledgeable about how to use R (including loading packages from Github), I don't see any alternative at present to using 'baydem'.

One further, minor note: the author makes the assertion that "KDE is a non-parametric method and does not require assumptions of a particular underlying distribution, unlike parametric Bayesian approaches or summed probability distributions." Two thirds of this statement are true, but it is incorrect to say that summed probability distributions require assumptions of a particular underlying distribution. When producing SPDs, one can but need not make any such assumptions, and in fact in most cases users of SPDs do not do so.

References

Bouveyron, C., Celeux, G., Murphy, T. B., and Raftery, A. E. 2019. Model-based Clustering and Classification for Data Science With Applications in R. Cambridge University Press, Cambridge.

Bronk Ramsey, C. 2017. Methods for summarizing radiocarbon datasets. Radiocarbon 59(6): 1809-1833.

Fraley, C., and Raftery, A.E. 2002. Model-Based Clustering, Discriminant Analysis, and Density Estimation. Journal of the American Statistical Association 97(458): 611-631.

Fraley, C., and Raftery, A.E. 2007. Bayesian Regularization for Normal Mixture Estimation and Model-Based Clustering. Journal of Classification 24: 155-188.

Gelman, A., Carlin, J. B., Stern, H.S., Dunson, D. B., Vehtari, A., and Rubin, D.B. 2013. Bayesian Data Analysis, 3rd ed. CRC Press, Boca Raton.

Murphy, K. P. 2012. Machine Learning: A Probabilistic Perspective. The MIT Press, Cambridge, MA.

Price, M.H., Capriles, J. M., Hoggarth, J. A., Bocinsky, R. K., Ebert, C. E., and Jones, J. H. 2021. End-to-end Bayesian analysis for summarizing sets of radiocarbon dates. Journal of Archaeological Science 135: 105473.

Silverman, B.W. 1986. Density Estimation for Statistics and Data Analysis. Chapman & Hall/CRC, Boca Raton.

Simonoff, J.S. 1996. Smoothing Methods in Statistics. Springer, New York.

6. PLOS authors have the option to publish the peer review history of their article (what does this mean?). If published, this will include your full peer review and any attached files.

Reviewer #1: No

Reviewer #2: **Yes: **William Brown

---

## [Author Response · Author response to Decision Letter 0]

1 Dec 2022

Response to Reviewer #1:

Reviewer #1: The author has compiled 538 radiocarbon dates from 142 archaeological sites across northwestern Washington State. The dates are grouped into three regions in order to compare demographic trends between them and also with other demographic models existing to the north, in British Columbia, Canada. Utilizing and comparing marine shell dates was also an objective of this study.

The compilation and analysis of these radiocarbon dates has the potential to be a significant contribution, but this study, as is, provides only a cursory and uncritical summary of the data, description of the results, and discussion of demographic trends. More contextual information is necessary for anything beyond the most basic modelling of demography. As it is, a reader would have a challenging time trying to evaluate the appropriateness of the research questions or the validity of the interpretations because no background information in offered on the nature of the c14 dated sites, or the known settlement history of the region. Readers unfamiliar with the region would reasonably wonder how the analysis of the compiled c14 dates relates to earlier research in the region, and advances understanding of key anthropological or historical questions.

A background section has been added to specifically note research questions in the region and their role in broader anthropological discussions. Additionally, it emphasizes that this paper is intended to provide a data synthesis similar to that done in BC and notes a model in BC to contrast large scale patterns with (Clark 2013).

Another concern with this study is that there is no effort to look at the settlement patterning over time, to detect if the types of sites or the distribution of sites changed in relation to one another, the environment, new technologies and settlement forms/size, or demographic trends. There is no discussion of factors that might help to explain some of the modeled ups and down in the KDE.

The background section should help address this concern. However settlement pattern analysis is not the focus of this particular paper nor is an analysis of site size and scope. An discussion of residential site sizes is in Rorabaugh 2017. Stein (2003) provides a seminal discussion of shell midden site formation that can also make behavioral interpretations based on site size alone problematic. 

Additional language has been added to link the results to the background section but also to highlight that the processes are not simply tied to population pressure or resource intensification.

I am not qualified to determine if the marine reservoir corrections have been applied appropriately, but more details would be helpful, and critical analysis of the methods should also be provided considering the ongoing uncertainty and regional variability with using shell dates.

Daniels (2009), Osterkamp et al. (2014), and Deo et al. (2004) provide more in depth discussions of marine shell calibrations. Which is beyond the scope of this paper beyond emphasizing that the error term introduction from these dates is appropriate for meso scale analyses.

The KDE seem to indicate some interesting trends that are generally consistent with those of other studies, but again, there is not sufficiently detailed results to know what the significance of this might be. Where the models indicate differing trends, there is also not discussion of why this might be.

The significance is that the US-Canadian border is an artificial barrier dividing academic research of a cultural area with shared history. Additional sentences have been added pertaining to an alternative model (Clark 2013) and that the dataset does not correspond with the idea of a core Fraser Delta and periphery comprised of the sample.

This manuscript contains many sentence fragments (including the first line of the introduction) and some repetition. Other sections are too brief to be clear to the reader. A background section that might provide useful context for interpretations and discussion is missing.

A new background section has been added that should address these concerns. The manuscript has been gone through for sentence clarity.

Altogether, I'd like to see a research paper along these lines published, but would expect it to be thoroughly re-worked, from the intro to the conclusion.

Based on the comments it is unclear what another paper would look like that they are requesting except for a quite different settlement pattern analysis which is not the thrust of this study. 

However, a background section and conclusions highlighting the specific findings situated in that background context have been added which should address these concerns within the scope of the aim of this study.

From reviewer #2 and #1s comments some additional areas for future work have been added which may be the paper that reviewer #1 may be envisioning but, again, is beyond the scope of this particular study. The settlement pattern analyses requested would be a different undertaking involving assessing spatial data quality from various sources but I would point the reviewer to the Taylor et al. article and Rorabaugh 2017 chapter 2 for discussions on household change through time and San Juan island site settlement pattern change through time. 

Response to Reviewer #2

Reviewer #2: This paper updates Taylor et al.'s 2011 paper, exploring population growth dynamics of the Pacific Northwest using the temporal distributions of radiocarbon datasets as proxies for the increasing and decreasing abundance of the archaeological record over time, i.e. the so-called "dates as data" approach. The present manuscript expands on Taylor et al.'s previous work in two noteworthy ways: first, it expands the geographic scope and consequently the sample size of the radiocarbon datasets used by Taylor et al., including adding inland sites from Washington State. Second, it replaces the summed probability distribution (SPD) approach used by Taylor et al. for describing temporal distributions of data points with a kernel density estimation (KDE) approach. Specifically, the author applies the KDE functions introduced to OxCal by Bronk Ramsey (2017), which have the appeal of being (relatively) easily applied by OxCal users without requiring any extensive programming on the user's part.

In general, I think this manuscript makes a novel and nontrivial contribution to our demographic-archaeological knowledge of the Pacific Northwest, and I do hope that it will be published for this reason. I also think that OxCal's KDE functionality supersedes both the SPD approach and other "composite kernel density estimation" alternatives suggested by others reviewed in the submission (e.g. Brown 2017). There is one alternative approach on offer---finite mixture modeling---that is founded on even strong theoretical and methodological principles than OxCal's KDE approach, which I discuss further below, but I will leave it to the author to decide whether the added effort to pursue finite mixture models over a KDE models is worth it; OxCal's KDE functions are still great tools.

Agreed with finite mixing model and would be interested in working with Reviewer #2 on such an approach as a future step.

One thing I would strongly recommend is that the author give the manuscript a new round of revisions with fresh eyes especially for broken grammar. Most of the paper is clearly written, but I did encounter a handful of sentence fragments, primarily sentence subjects lacking their predicates. Make sure every sentence ends.

Manuscript has been checked for incomplete sentences and a pass for incomplete grammar.

Grammar aside, I do have one important critique of the manuscript that I think will need to be addressed before it is published: the author's characterization of kernel density estimation (KDE) as a Bayesian method is incorrect. In fact, KDE is not readily available to standard likelihood based approaches to model-fitting---whether maximum likelihood estimation (MLE; Silverman 1986: 25-26, 110-119; Simonoff 1996: 64-70) or Bayesian inference (Bouveyron et al. 2019: 92-93)---because tuning the KDE model's single free parameter ("bandwidth") in a likelihood-based way leads to the statistical problem of degeneracy: the likelihood function approaches infinity as the strictly positive bandwidth parameter approaches 0. In the case of KDE and other finite mixture models (FMMs), this degeneracy problem sometimes goes under the label "the collapsing variance problem" (Murphy 2012: 357-359), and it is unavoidable in the case of KDE. While it is true in general that OxCal comprises a diverse set of tools for radiocarbon data analysis that are Bayesian in a strict sense, this is not the case for OxCal's recently added KDE operations, a fact about which Bronk Ramsey (2017) is quite explicit. Instead, in order to operationalize KDE, OxCal's almost-but-not-quite-Bayesian approach replaces the proper likelihood function required by Bayes' Rule with a leave-one-out predictive probability function (Bronk Ramsey 2017: Eqs. 5 and 9). (A similar approach has been applied to MLE approaches to KDE; Silverman 1986: 52-55.) On one hand, this substitution "works" because the out-of-sample predictive probability function behaves in a manner similar to likelihood functions, increasing and decreasing over the parameter space depending on how suitable different parameter values are for the data under analysis. On the other hand, these out-of-sample predictive probabilities still should not be mistaken for likelihoods in the strict sense, so their substitution is of questionable theoretical merit. In my opinion, this choice is especially problematic for Bayesian inference because this substitution sacrifices the logical axioms that give this inferential framework, especially Bayes' Rule, its principled foundations. Nevertheless, a certain amount of pragmatic justification does exist for their use as a hack when proper likelihoods just won't work (Silverman 1986: 52-55), so I wouldn't insist that we abandon OxCal's KDE functions altogether. Even so, they are not Bayesian.

In any case, the author will need to make one of two changes to address and resolve this issue:

(1) The simpler solution would be editorial: remove or qualify any characterization of KDE as Bayesian, including in the paper title, key words, abstract, and main body of the text. This would require no new analysis on the author's part but merely a more accurate characterization of what the author has already done.

Approach #1 was taken, and a more accurate characterization of the work done has been included. Approach #2 would be an excellent approach to start with a smaller portion of the dataset such as 45WH1 Cherry Point or the Woodstock Farm site where there are strong 

(2) Alternatively, if the author really wishes to do something that is truly Bayesian, he should consider a new round of analysis, replacing his KDE models with FMMs that can be approached in a Bayesian way. As it turns out, KDE models are an extreme case of FMMs, in which there are as many mixture components as there are data points and in which all kernels or mixture components share the same bandwidth. However, unlike KDE models specifically, FMMs more generally *are* available to Bayesian inference in the strict sense, particularly if there are fewer kernels/mixture components than there are data points. The collapsing variance problem mentioned earlier is still a possibility particularly for FMMs with large numbers of mixture components, but this problem can be mitigated in a Bayesian framework using regularizing priors (Fraley and Raftery 2007; Bouveyron 2019: 92-95) and/or posterior maximization rather than likelihood maximization (Murphy 2012: 357-359). Additionally, using FMMs with relatively few mixture components usually out-performs KDEs when applied as density estimates to the same data, particularly if different kernels/mixture components are allowed to have different bandwidths (Fraley and Raftery 2002; Gelman et al. 2013: 535). Unfortunately, OxCal has not yet implemented functions for finite mixture modeling. To my knowledge, the only prepackaged program capable of fitting such models is the 'baydem' package for R described by Price et al. (2021). If the author wishes to apply a truly Bayesian approach to the task at hand and is knowledgeable about how to use R (including loading packages from Github), I don't see any alternative at present to using 'baydem'.

One further, minor note: the author makes the assertion that "KDE is a non-parametric method and does not require assumptions of a particular underlying distribution, unlike parametric Bayesian approaches or summed probability distributions." Two thirds of this statement are true, but it is incorrect to say that summed probability distributions require assumptions of a particular underlying distribution. When producing SPDs, one can but need not make any such assumptions, and in fact in most cases users of SPDs do not do so.

The sentence has been corrected to remove the SPD portion to be accurate.

---

## [Decision Letter · Decision Letter 1]

2 Jan 2023

PONE-D-22-25181R1Assessing Population Dynamics in the Central Salish SeaPLOS ONE

Dear Dr. Rorabaugh,

Thank you for submitting your manuscript to PLOS ONE. After careful consideration, we feel that it has merit but does not fully meet PLOS ONE’s publication criteria as it currently stands. Therefore, we invite you to submit a revised version of the manuscript that addresses the points raised during the review process. Per the two reviews, your manuscript needs more work before it will be acceptable for publications. You will need to take all of the comments and suggestions into account when making revisions.

We look forward to receiving your revised manuscript.

Kind regards,

John P. Hart, Ph.D.

Academic Editor

PLOS ONE

Reviewers' comments:

Reviewer's Responses to Questions

**Comments to the Author**

1. If the authors have adequately addressed your comments raised in a previous round of review and you feel that this manuscript is now acceptable for publication, you may indicate that here to bypass the “Comments to the Author” section, enter your conflict of interest statement in the “Confidential to Editor” section, and submit your "Accept" recommendation.

Reviewer #1: (No Response)

Reviewer #2: (No Response)

2. Is the manuscript technically sound, and do the data support the conclusions?

Reviewer #1: Partly

Reviewer #2: Yes

3. Has the statistical analysis been performed appropriately and rigorously? 

Reviewer #1: Yes

Reviewer #2: Yes

4. Have the authors made all data underlying the findings in their manuscript fully available?

Reviewer #1: Yes

Reviewer #2: (No Response)

5. Is the manuscript presented in an intelligible fashion and written in standard English?

Reviewer #1: Yes

Reviewer #2: No

6. Review Comments to the Author

Reviewer #1: The author’s removal of the term “settlement” from the title of the manuscript helps to clarify the focus of this paper on population, but the rest of the edits do not address the principal issues that I identified.

As originally noted, the manuscript (from the first paragraph of the introduction) pre-supposes that readers will have an understanding of the paleo demography of the San Juan Islands and Salish Sea. I think that more work is required to demonstrate how this study contributes to the study of population dynamics in the region.

In the second paragraph of the introduction the author outlines the geographic divisions for analysis, but no rationale is provided. I wonder why the radiocarbon dates from the San Juan islands are analyzed separately from the adjoining mainland? Based on historical and cultural factors the data could just as easily be grouped. I think that the geographic divisions make intuitive sense, but this could be elaborated on, especially because the relationship between the islands and the mainland seems to be importation to some of the conclusions (ie. an influx of people from the mainland to the islands).

The author added a general overview of the region with an emphasis on hereditary social inequality. It’s not clear how this all ties in with the research questions. I would have expected context relating to populations/demography, archaeological perspectives on the growth and distribution of various localized or regional populations. Or perhaps on population movements, since this is proposed in the study. Some parts of this overview would also need to be expanded on to provide adequate background information for non-regional archaeologists.

The research questions also pre-suppose that the reader has existing familiarity with the region (ie., Clark’s 2013 idea that the Fraser Delta was a regional core) and previous San Juan Island research in particular.

the first couple lines of the second paragraph of the research questions section seem unrelated to the rest of the paper.

Methodology: It’s not clear how complete this compilation of radiocarbon dates is? This would be useful to know. Also when was the data compiled (ie. how up-to-date is it)?

Figure 3. Many of the sites look like they are in the middle of the ocean. Must be a mapping error.

Map figures: It seems important to represent the riverways, since nearly all the mainland sites are oriented to them.

KDE plot figures. It’s hard for the reader to evaluate the differences between Marine vs. Terrestrial data or even the regions because the timescales on the plots are different. Ideally, they could be plotted together, with variances statistically represented.

The results sections are all very brief and could be described more clearly so that the reader can get a better sense of how the KDE plots might relate to population dynamics.

In the discussion the author seems to suggest that the San Juan population dynamics reflect an “influx” of people around 2,600 – 2,200 cal BP and notes that it is too simplistic to focus on resource intensification as either a driving force or result of greater populations. Because there is no discussion of demography in this manuscript, it does not seem warranted to conclude that changes in the KDE were the result of an influx of people rather than in-place demographic growth (which I would assume is the viewpoint of the Indigenous communities). The author’s critques of other models (Croes and Hackenburger 1988 and Clark 2013) are not clearly refuted with the presented data or described in a way that can be evaluated. Also, in effect, the author seems to be describing the same sort of expansion idea, only from the Skagit River/delta to the San Juan Islands.

Overall the discussion should be developed and expanded so that readers can understand the contributions of the study, and how it advances other recent explorations of this same topic. One study that the author should consider is the recent Prentiss et al. 2022 study on population dynamics comparing the middle Fraser and lower Fraser regions (Divergent population dynamics in the middle to late Holocene lower Fraser valley and mid-Fraser canyon, British Columbia).

Issues with punctuation and spelling persist. There is a grammatical issue with the first sentence of the abstract.

Reviewer #2: I am fully satisfied that all of the theoretical and substantive issues I raised in my first review have been addressed. That being said, I noted the need for some further copy editing in my initial review as well, and unfortunately some important editorial problems persist in the manuscript, including in the abstract. I will list those that I think undermine the reader experience most, in order of appearance. Because the manuscript lacks page or line numbers, I will have to use the section name and paragraph number within each section to orient the author for the purpose of correction:

Abstract: The grammar is broken in the sentence "Recent developments in radiocarbon dating have enabled archaeologists to re-examination of question of population dynamism in the Salish Sea". I suggest revising "re-examiniation of question" to "reexamine the question"

Abstract: The words "that the" should be dropped from the following passage: "The expanded sample suggests that the patterns of population growth from 3200-2800 cal BP ...". Also, consider changing "patterns" to "a pattern"

Section: Introduction, Para. 2: "beingutilized" should be "being utilized"

Section: Research questions, para. 1: "If the case" should be "If this is the case"

Section: Database, para. 1: This is a run-on sentence: "An aim of this study was to critically assess uncertainties in marine dates, shell and bone dates with marine diet contributions were included." I think the logical connector "so" needs to be added after the comma. Alternatively, change the comma to a period and begin the new sentence that follows with "Consequently."

Section: Kernel density estimation, para. 2: typo: "methofd" should be "method"

Section: Discussion, para. 6: Revisit the grammar and punctuation of this sentence: "While an FMM study of individual sites is more in line with previous site-level radiocarbon analyses previously done in the region, can provide new insights to specific site histories and would be complementary to a settlement pattern analysis approach."

Additionally, I disagree with the implication that an FMM would be more inline with intra-site analysis than inter-site analyses. On one hand, it is true that finite mixture modeling is very often used as a method for model-based cluster analysis (e.g., see Bouveyrone et al. 2019 cited in my first round of review). Consequently, I would agree with the author that FMMs would be well-suited to the task of disentangling palimpsests of radiocarbon data at an intra-site level. On the other hand, FMMs can also be used simply as nonparametric density estimators. In fact, as I noted in my first review, FMM includes KDE as a special, extreme case. In this case, the mixture components are not interpreted as representing latent subpopulations in a mixed population but merely as tools for inducing flexible distributional structures in the density estimators. Consequently, FMMs are just as well suited for modeling regional dynamics in settlement patterns and demographic change as they are for disentangling latent clusters in mixed populations. This is actually the application that Price has in mind in developing the R 'bayesdem' package.

Section: Discussion, para. 7: This sentence needs some serious editorial reworking: "2. The marine and terrestrial sample lacks some finer structure seen in terrestrial only sample but broad patterns relatively consistent, ...". Perhaps it could be revised to something like "The combined marine and terrestrial sample lacks the fine distributional structures seen in the terrestrial-only sample, but broad patterns are consistent between the two, ..."

Section: Discussion, para. 7: The word "be" is missing between "not" and "problematic" in the following passage: "3. While shell date error terms may obscure fine structure, reservoir corrections themselves may not problematic."

7. PLOS authors have the option to publish the peer review history of their article (what does this mean?). If published, this will include your full peer review and any attached files.

Reviewer #1: No

Reviewer #2: **Yes: **William A. Brown

---

## [Author Response · Author response to Decision Letter 1]

15 Feb 2023

Response to Reviewers:

1. If the authors have adequately addressed your comments raised in a previous round of review and you feel that this manuscript is now acceptable for publication, you may indicate that here to bypass the “Comments to the Author” section, enter your conflict of interest statement in the “Confidential to Editor” section, and submit your "Accept" recommendation.

Reviewer #1: (No Response)

Reviewer #2: (No Response)

2. Is the manuscript technically sound, and do the data support the conclusions?

Reviewer #1: Partly

Additional edits have been made to better connect the requested background sections and discussion as working hypotheses of the data including connecting the ethnographic discussions more explicitly with population dynamics and the addition of corroborating oral tradition. However, as reviewer #2 is a subject matter expert regarding the methods, if there are technical issues with the data, statistics, and presentation, I defer to them on those matters 

Reviewer #2: Yes

3. Has the statistical analysis been performed appropriately and rigorously?

Reviewer #1: Yes

Reviewer #2: Yes

4. Have the authors made all data underlying the findings in their manuscript fully available?

Reviewer #1: Yes

Reviewer #2: (No Response)

5. Is the manuscript presented in an intelligible fashion and written in standard English?

Reviewer #1: Yes

Reviewer #2: No

All suggested edits by reviewer #2 have been incorporated and additional copy editing for sections altered to address reviewer #1.

6. Review Comments to the Author

Reviewer #1: The author’s removal of the term “settlement” from the title of the manuscript helps to clarify the focus of this paper on population, but the rest of the edits do not address the principal issues that I identified.

As originally noted, the manuscript (from the first paragraph of the introduction) pre-supposes that readers will have an understanding of the paleo demography of the San Juan Islands and Salish Sea. I think that more work is required to demonstrate how this study contributes to the study of population dynamics in the region.

The paper now explicitly defines the Salish Sea, and I have included that this is the first examination of dates for this region beyond the aforementioned San Juan archaeological project. This study expands on that work with additional data, and provides the first synthesis of the non-island dates from Northwestern Washington state.

In the second paragraph of the introduction the author outlines the geographic divisions for analysis, but no rationale is provided. I wonder why the radiocarbon dates from the San Juan islands are analyzed separately from the adjoining mainland? Based on historical and cultural factors the data could just as easily be grouped. I think that the geographic divisions make intuitive sense, but this could be elaborated on, especially because the relationship between the islands and the mainland seems to be importation to some of the conclusions (ie. an influx of people from the mainland to the islands).

More information has been added on the specific region research histories, how they inform the regional divisions, and why this particular study has been undertaken.

To address the comments on the relationship between the islands and the mainland, oral history accounts of the arrival to the San Juan Islands, and leaving the islands to the mainland have been added. The data appear to support these oral traditions. This has been added to the discussion.

The author added a general overview of the region with an emphasis on hereditary social inequality. It’s not clear how this all ties in with the research questions. I would have expected context relating to populations/demography, archaeological perspectives on the growth and distribution of various localized or regional populations. Or perhaps on population movements, since this is proposed in the study. Some parts of this overview would also need to be expanded on to provide adequate background information for non-regional archaeologists.

The research questions also pre-suppose that the reader has existing familiarity with the region (ie., Clark’s 2013 idea that the Fraser Delta was a regional core) and previous San Juan Island research in particular.

Providing a full literature review of the SJAP would be beyond the scope of a typical paper for this journal. However the earlier paragraphs explicitly discussing research history should address this. Additional context has been added for Clark for those not familiar with his sequence and culture-history based arguments, but described without the specific sequence building and references have been added to critiques levelled.

the first couple lines of the second paragraph of the research questions section seem unrelated to the rest of the paper.

Added connecting sentence about concerns raised regarding erosional effects on the study. Clarified that the erosional effects would not change overall trends in the study.

Methodology: It’s not clear how complete this compilation of radiocarbon dates is? This would be useful to know. Also when was the data compiled (ie. how up-to-date is it)?

Added additional clarification of the date range 1969-2022, and that the dataset includes all MAs, dissertations, gray literature, and an unpublished WWU database.

Figure 3. Many of the sites look like they are in the middle of the ocean. Must be a mapping error.

Figures have been adjusted to have a higher resolution open access shoreline from the state of Washington that includes minor islands.

Map figures: It seems important to represent the riverways, since nearly all the mainland sites are oriented to them.

Figures have been updated to include riverways from open access state of Washington data.

KDE plot figures. It’s hard for the reader to evaluate the differences between Marine vs. Terrestrial data or even the regions because the timescales on the plots are different. Ideally, they could be plotted together, with variances statistically represented.

Timescales for the figures have been adjusted, new figures have been made for each region combining the sets of unmodeled date probabilities and posterior probabilities combining the plots. The raw model data and posterior probabilities do not include variances. The reviewer is likely referring to data display from composite KDE which is not employed here which would be a potential next step, although an FMM model of specific sites is the immediate next direction. 

The final figures included posterior probabilities directly comparing each region, but have been adjusted with new scales.

The results sections are all very brief and could be described more clearly so that the reader can get a better sense of how the KDE plots might relate to population dynamics.

Additional text has been added for how the posterior probabilities relate to overall population trends. Including lower population densities in the interior throughout time and how island population dynamics reflect Lummi oral traditions, at the recommendation of community members.

In the discussion the author seems to suggest that the San Juan population dynamics reflect an “influx” of people around 2,600 – 2,200 cal BP and notes that it is too simplistic to focus on resource intensification as either a driving force or result of greater populations. Because there is no discussion of demography in this manuscript, it does not seem warranted to conclude that changes in the KDE were the result of an influx of people rather than in-place demographic growth (which I would assume is the viewpoint of the Indigenous communities). 

Specific oral history narratives regarding Lummi population movement have been added. However, the primary thrust has been in situ population growth and dynamism with population movement due to the nature of the marriage and social networks which the background section included but additional information tied to population dynamics was added. The addition of these oral narratives for interpretive context of the results reflects the viewpoint of the Indigenous community.

The author’s critques of other models (Croes and Hackenburger 1988 and Clark 2013) are not clearly refuted with the presented data or described in a way that can be evaluated. Also, in effect, the author seems to be describing the same sort of expansion idea, only from the Skagit River/delta to the San Juan Islands.

It was is not the point of this article to refute Croes and Hackenberger's model of general in situ demographic increase but instead noting that the literature has moved beyond using that as a social prime mover based on faunal evidence. However, the critiques of that and Clark are cited with additional context. The main thrust of this is providing a regional synthesis of dates from the gray literature and working hypotheses (now situated in Lummi oral tradition) for the trends observed.

Additionally, Lummi oral traditions note expansion from north to the Islands, and subsequent movement from the islands to the mainland. The direction of expansion is from the San Juans to the Skagit River, and appears to be reflected in the data. Similarly the discussion on hereditary inequality is important to provide context for “voting with your feet” which is a prime driver for population dynamism in the area but additional language has been added to make this explicit. 

Overall the discussion should be developed and expanded so that readers can understand the contributions of the study, and how it advances other recent explorations of this same topic. One study that the author should consider is the recent Prentiss et al. 2022 study on population dynamics comparing the middle Fraser and lower Fraser regions (Divergent population dynamics in the middle to late Holocene lower Fraser valley and mid-Fraser canyon, British Columbia).

The Prentiss study is now cited, but the state of research in this particular area is less developed outside of the SJAP than the ongoing efforts by Prentiss and colleagues. 

Issues with punctuation and spelling persist. There is a grammatical issue with the first sentence of the abstract.

Fixed

Reviewer #2: I am fully satisfied that all of the theoretical and substantive issues I raised in my first review have been addressed. That being said, I noted the need for some further copy editing in my initial review as well, and unfortunately some important editorial problems persist in the manuscript, including in the abstract. I will list those that I think undermine the reader experience most, in order of appearance. Because the manuscript lacks page or line numbers, I will have to use the section name and paragraph number within each section to orient the author for the purpose of correction:

Noted issues have been fixed and the manuscript was ran through additional copyediting post edits made to address reviewer #1.

Abstract: The grammar is broken in the sentence "Recent developments in radiocarbon dating have enabled archaeologists to re-examination of question of population dynamism in the Salish Sea". I suggest revising "re-examiniation of question" to "reexamine the question"

Fixed

Abstract: The words "that the" should be dropped from the following passage: "The expanded sample suggests that the patterns of population growth from 3200-2800 cal BP ...". Also, consider changing "patterns" to "a pattern"

Fixed

Section: Introduction, Para. 2: "beingutilized" should be "being utilized"

Fixed

Section: Research questions, para. 1: "If the case" should be "If this is the case"

Fixed

Section: Database, para. 1: This is a run-on sentence: "An aim of this study was to critically assess uncertainties in marine dates, shell and bone dates with marine diet contributions were included." I think the logical connector "so" needs to be added after the comma. Alternatively, change the comma to a period and begin the new sentence that follows with "Consequently."

Fixed

Section: Kernel density estimation, para. 2: typo: "methofd" should be "method"

Fixed

Section: Discussion, para. 6: Revisit the grammar and punctuation of this sentence: "While an FMM study of individual sites is more in line with previous site-level radiocarbon analyses previously done in the region, can provide new insights to specific site histories and would be complementary to a settlement pattern analysis approach."

Fixed

Additionally, I disagree with the implication that an FMM would be more inline with intra-site analysis than inter-site analyses. On one hand, it is true that finite mixture modeling is very often used as a method for model-based cluster analysis (e.g., see Bouveyrone et al. 2019 cited in my first round of review). Consequently, I would agree with the author that FMMs would be well-suited to the task of disentangling palimpsests of radiocarbon data at an intra-site level. On the other hand, FMMs can also be used simply as nonparametric density estimators. In fact, as I noted in my first review, FMM includes KDE as a special, extreme case. In this case, the mixture components are not interpreted as representing latent subpopulations in a mixed population but merely as tools for inducing flexible distributional structures in the density estimators. Consequently, FMMs are just as well suited for modeling regional dynamics in settlement patterns and demographic change as they are for disentangling latent clusters in mixed populations. This is actually the application that Price has in mind in developing the R 'bayesdem' package.

The sentence wasn't meant to imply that FMM was more appropriate for disentangling mixed populations but that it's best use for this dataset would be that type of approach. I've added a clarification sentence that it is not exclusive for the smaller scale and just as suitable.

Section: Discussion, para. 7: This sentence needs some serious editorial reworking: "2. The marine and terrestrial sample lacks some finer structure seen in terrestrial only sample but broad patterns relatively consistent, ...". Perhaps it could be revised to something like "The combined marine and terrestrial sample lacks the fine distributional structures seen in the terrestrial-only sample, but broad patterns are consistent between the two, ..."

Adjusted to recommended language.

Section: Discussion, para. 7: The word "be" is missing between "not" and "problematic" in the following passage: "3. While shell date error terms may obscure fine structure, reservoir corrections themselves may not problematic."

Fixed

7. PLOS authors have the option to publish the peer review history of their article (what does this mean?). If published, this will include your full peer review and any attached files.

Do you want your identity to be public for this peer review? For information about this choice, including consent withdrawal, please see our Privacy Policy.

Reviewer #1: No

Reviewer #2: Yes: William A. Brown

---

## [Decision Letter · Decision Letter 2]

27 Mar 2023

PONE-D-22-25181R2Assessing Population Dynamics in the Central Salish SeaPLOS ONE

Dear Dr. Rorabaugh,

Thank you for submitting your manuscript to PLOS ONE. After careful consideration, we feel that it has merit but does not fully meet PLOS ONE’s publication criteria as it currently stands. Therefore, we invite you to submit a revised version of the manuscript that addresses the points raised during the review process. Thank you for making the revisions to the manuscript. Both reviewers are satisfied that the revisions have addressed their concerns. Reviewer 1 identifies some minor issues that you should consider when making your final revisions prior to publication.

We look forward to receiving your revised manuscript.

Kind regards,

John P. Hart, Ph.D.

Academic Editor

PLOS ONE

Journal Requirements:

Reviewers' comments:

Reviewer's Responses to Questions

**Comments to the Author**

1. If the authors have adequately addressed your comments raised in a previous round of review and you feel that this manuscript is now acceptable for publication, you may indicate that here to bypass the “Comments to the Author” section, enter your conflict of interest statement in the “Confidential to Editor” section, and submit your "Accept" recommendation.

Reviewer #1: All comments have been addressed

Reviewer #2: All comments have been addressed

2. Is the manuscript technically sound, and do the data support the conclusions?

Reviewer #1: Partly

Reviewer #2: Yes

3. Has the statistical analysis been performed appropriately and rigorously? 

Reviewer #1: Yes

Reviewer #2: Yes

4. Have the authors made all data underlying the findings in their manuscript fully available?

Reviewer #1: Yes

Reviewer #2: Yes

5. Is the manuscript presented in an intelligible fashion and written in standard English?

Reviewer #1: (No Response)

Reviewer #2: Yes

6. Review Comments to the Author

Reviewer #1: Overall, I find that this manuscript has improved considerably from the original submission, but there are still a few changes/additions that would ideally be addressed before publication.

One issue is that there is no discussion on population growth overall, starting with radiocarbon evidence for people living in inland, then on the coast, then on the islands. I’m not clear why there is no real mention of the broader trends as indicated by the radiocarbon dates. The discussion focuses-in on only a couple peaks (mostly related to the Islands). This is fine of course, but it’s not explained why some peaks are emphasized more than others, or over the general trends.

At one point the author notes that there is in-situ population growth in NW coastal Washington between 3,200 – 2,800 cal BP. People were living in this area long before this time though…was this earlier time characterized by something other than in-situ growth? I think that this is only confusing because there is not a more general review discussion on longer-term population trends.

In the San Juan results section the first peak is described as being between 2,400 – 2,200 cal BP. In the discussion it’s reported as starting at 2,600 cal BP. Which is more accurate?

So, even though people have been living on the San Juan islands since before 3,000 years ago, the argument is that the First Ancestor of the Lummi people may have arrived around 2,600/2,400 cal BP because there is an increase in population at this time?

I also wonder about the word choice of “influx” to describe this significant event in tribal history. Influx makes it sound like the First Ancestor is coming from somewhere nearby (like the mainland, as was suggested in the second version). Typically though, these oral histories are explicit about the ancestors coming from the sky, and that they are the original settlers in a new area.

The figures are better than before, but could still be improved with place names (ie., coastal and riverine water bodies) to help orient the reader.

Figure 1. “Stolo” is the only name provided for a cultural group on the Canadian side of the map, and it’s use is inaccurate. I suggest sticking with linguistic groupings at this geographic scale.

Reviewer #2: (No Response)

7. PLOS authors have the option to publish the peer review history of their article (what does this mean?). If published, this will include your full peer review and any attached files.

Reviewer #1: No

Reviewer #2: **Yes: **William Brown

---

## [Author Response · Author response to Decision Letter 2]

11 Apr 2023

Response to Reviewers:

1. If the authors have adequately addressed your comments raised in a previous round of review and you feel that this manuscript is now acceptable for publication, you may indicate that here to bypass the “Comments to the Author” section, enter your conflict of interest statement in the “Confidential to Editor” section, and submit your "Accept" recommendation.

Reviewer #1: All comments have been addressed

Reviewer #2: All comments have been addressed

2. Is the manuscript technically sound, and do the data support the conclusions?

Reviewer #1: Partly

Reviewer #2: Yes

3. Has the statistical analysis been performed appropriately and rigorously?

Reviewer #1: Yes

Reviewer #2: Yes

4. Have the authors made all data underlying the findings in their manuscript fully available?

Reviewer #1: Yes

Reviewer #2: Yes

5. Is the manuscript presented in an intelligible fashion and written in standard English?

Reviewer #1: (No Response)

Reviewer #2: Yes

6. Review Comments to the Author

Reviewer #1: Overall, I find that this manuscript has improved considerably from the original submission, but there are still a few changes/additions that would ideally be addressed before publication.

One issue is that there is no discussion on population growth overall, starting with radiocarbon evidence for people living in inland, then on the coast, then on the islands. I’m not clear why there is no real mention of the broader trends as indicated by the radiocarbon dates. The discussion focuses-in on only a couple peaks (mostly related to the Islands). This is fine of course, but it’s not explained why some peaks are emphasized more than others, or over the general trends.

An additional paragraph outlining the demographic work done in the 1980s and 1990s has been added and a new figure with the broader trends for the entire Salish Sea. Specific callouts to the emphasized peaks diverging from the "regional demographic model" have been added. An additional sentence in the discussion about multi-scalar analyses being important that combining current data sets shows a general curve consistent with the older models, but obscuring regional structure, has also been added. 

At one point the author notes that there is in-situ population growth in NW coastal Washington between 3,200 – 2,800 cal BP. People were living in this area long before this time though…was this earlier time characterized by something other than in-situ growth? I think that this is only confusing because there is not a more general review discussion on longer-term population trends.

There is no evidence of earlier population growth not being in situ in that region, the sentence has been updated to state "continuing in situ population growth" for clarity

In the San Juan results section the first peak is described as being between 2,400 – 2,200 cal BP. 

In the discussion it’s reported as starting at 2,600 cal BP. Which is more accurate?

The discussion and abstract are correct in that it is 2,600 cal BP to 2,200 cal BP. 2400-2200 is the NW WA sample the text has been adjusted accordingly.

So, even though people have been living on the San Juan islands since before 3,000 years ago, the argument is that the First Ancestor of the Lummi people may have arrived around 2,600/2,400 cal BP because there is an increase in population at this time?

That was not the intent of the statements and so text has been clarified that oral tradition supports the initial in situ population growth and the later movement to the mainland. If there was other large scale population movement from the mainland to the islands, beyond the noted village fission-fusion dynamics in the background and discussion, it is not supported at this time based on current evidence. The text has been adjusted to avoid readers making that misconception. The Island to mainland movement is supported by oral tradition and so the text has been adjusted further to clarify that.

I also wonder about the word choice of “influx” to describe this significant event in tribal history. Influx makes it sound like the First Ancestor is coming from somewhere nearby (like the mainland, as was suggested in the second version). Typically though, these oral histories are explicit about the ancestors coming from the sky, and that they are the original settlers in a new area.

In this case influx is appropriate for the Lummi to the mainland but the language in the abstract and discussion has been altered to "population growth" in the San Juans to be more consistent with oral traditions that the same ancestor in the San Juan Islands was an ancestor to populations in SE Vancouver Island. Additionally more information from place names has been added to note that these apical ancestors “appeared or came down” in a manner not unlike what Reviewer #1 suggests. 

The figures are better than before, but could still be improved with place names (ie., coastal and riverine water bodies) to help orient the reader.

Major riverine bodies of relevance (Nooksack River, Skagit River, Fraser River have been added. Strait of Juan de Fuca and Strait of Georgia also added.

Figure 1. “Stolo” is the only name provided for a cultural group on the Canadian side of the map, and it’s use is inaccurate. I suggest sticking with linguistic groupings at this geographic scale.

Additional referred to groups in text have been added to address this concern as well as the Musqueam to match with the Suttles reference.

Reviewer #2: (No Response)

7. PLOS authors have the option to publish the peer review history of their article (what does this mean?). If published, this will include your full peer review and any attached files.

Do you want your identity to be public for this peer review? For information about this choice, including consent withdrawal, please see our Privacy Policy.

Reviewer #1: No

Reviewer #2: Yes: William Brown

---

## [Editor Report · Decision Letter 3]

13 Apr 2023

Assessing Population Dynamics in the Central Salish Sea, Pacific Northwest Coast of North America

PONE-D-22-25181R3

Dear Dr. Rorabaugh,

We’re pleased to inform you that your manuscript has been judged scientifically suitable for publication and will be formally accepted for publication once it meets all outstanding technical requirements.

Kind regards,

John P. Hart, Ph.D.

Academic Editor

PLOS ONE
---

## [Editor Report · Acceptance letter]

18 Apr 2023

PONE-D-22-25181R3 

Assessing Population Dynamics in the Central Salish Sea, Pacific Northwest Coast of North America 

Dear Dr. Rorabaugh:

I'm pleased to inform you that your manuscript has been deemed suitable for publication in PLOS ONE. Congratulations! Your manuscript is now with our production department. 

Kind regards, 

on behalf of

Dr. John P. Hart 

Academic Editor

PLOS ONE